# Direct observation of dynamic protein interactions involving human microtubules using solid-state NMR spectroscopy

Yanzhang Luo [1,7], ShengQi Xiang [1,2,7], Peter Jan Hooikaas [3], Laura van Bezouwen[4], A.S. Jijumon [5,6], Carsten Janke [5,6], Friedrich Förster [4], Anna Akhmanova [3]* & Marc Baldus [1]*

Microtubules are important components of the eukaryotic cytoskeleton. Their structural organization is regulated by nucleotide binding and many microtubule-associated proteins (MAPs). While cryo-EM and X-ray crystallography have provided detailed views of inter-actions between MAPs with the microtubule lattice, little is known about how MAPs and their intrinsically disordered regions interact with the dynamic microtubule surface. NMR carries the potential to directly probe such interactions but so far has been precluded by the low tubulin yield. We present a protocol to produce [$^{13}$C, $^{15}$N]-labeled, functional microtubules (MTs) from human cells for solid-state NMR studies. This approach allowed us to demonstrate that MAPs can differently modulate the fast time-scale dynamics of C-terminal tubulin tails, suggesting distinct interaction modes. Our results pave the way for in-depth NMR studies of protein dynamics involved in MT assembly and their interactions with other cellular components.

[1] NMR Spectroscopy, Bijvoet Center for Biomolecular Research, Utrecht University, Padualaan 8, 3584 CH Utrecht, The Netherlands. [2] MOE Key Lab for Membrane-less Organelles & Cellular Dynamics, School of Life Sciences, University of Science and Technology of China, 96 Jinzhai Road, Hefei 230026 Anhui, China. [3] Cell Biology, Neurobiology and Biophysics, Department of Biology, Faculty of Science, Utrecht University, Padualaan 8, 3584 CH Utrecht, The Netherlands. [4] Cryo-Electron Microscopy, Bijvoet Center for Biomolecular Research, Utrecht University, Padualaan 8, 3584 CH Utrecht, The Netherlands. [5] Institut Curie, PSL Research University, CNRS UMR3348, F-91405 Orsay, France. [6] Université Paris Sud, Université Paris-Saclay, CNRS UMR3348, F-91405 Orsay, France. [7] These authors contributed equally: Yanzhang Luo, ShengQi Xiang. *email: a.akhmanova@uu.nl; m.baldus@uu.nl

In eukaryotic cells, microtubules (MTs) are cytoskeletal polymers essential for many biological processes, including cell division, migration, polarization, and intracellular trafficking. MTs are assembled from α/β-tubulin heterodimers and are intrinsically polarized, with the highly dynamic, β-tubulin-exposed plus end, which rapidly switches between growth and shrinkage in a process termed "dynamic instability"[1]. The dynamic instability of MTs is caused by GTP binding and hydrolysis on tubulin dimers.

Many MT-associated proteins (MAPs) regulate MT structure and function by interacting with MT lattices and/or MT ends[2,3]. Cryo-electron microscopy (EM) has made substantial progress to elucidate the interaction between MAPs and the MT lattice[4,5], including the cryo-EM reconstruction of the MAP tau binding to MTs[6]. However, the direct observation of dynamic regions of tubulin, including the unstructured C-terminal tails that are critical for binding of different MT-associated proteins such as tau[6] has remained challenging due to their intrinsic flexibility. To model such tubulin regions computational approaches were used, for example in the context of tau binding to MTs[6] or, more recently, to determine the effect of α-tubulin acetylation on MT structure and stability[7]. Given the wide-spread relevance of protein dynamics for MT function and, in particular, of tubulin tails for cellular processes and human disorders[8], direct experimental insight into the dynamic interaction of MTs with MAPs is hence of significant interest.

Nuclear magnetic resonance (NMR) spectroscopy has been shown to provide unique structural insights into heterogeneous and dynamical systems at atomic resolution and on different time scales (see, e.g., refs. [9–12]). Previously, solid- and solution-state NMR have been used to study the interactions between MTs and isotope-labeled MAPs or small drugs[13–19]. However, these studies were restricted to studying ligands that could be isotope labeled using bacterial expression systems. Thus far, the direct study of functional MTs from human cells that carry NMR-active labeling was precluded due to insufficient protein amounts. In addition, solution-state NMR studies on intact MTs are hampered by protein size and insufficient molecular tumbling. In the following, we describe a solid-state NMR (ssNMR) approach to directly study the interaction of labeled MTs with their binding partners. In particular, this method enables us to probe the dynamics of intact MTs. Moreover, we examine the influence of two different MT-associated proteins upon the C-terminal tail dynamics. We investigate the binding of the CKK domain that is important for the minus-end recognition of the calmodulin-regulated spectrin-associated protein (CAMSAP)[15,19]. Furthermore, we examine the effect of the microtubule-binding domain (MTBD) of the MAP7 family of proteins, which play an important role in regulating kinesin-based intracellular transport and for which the structural details of MT binding are currently unknown[20–23]. Our results suggest that MAP-tubulin tail interactions can involve both fast-time scale interactions between mobile chains as well as slow-time scale binding/unbinding events within stable MT–MAP complexes.

## Results

**Functional MTs for ssNMR studies.** To obtain milligram quantities of human MTs we used HeLa S3 cells that not only grow as adherent culture but also in suspension, thereby increasing the efficiency of large-scale cultures[24]. In addition, HeLa S3 cells express only a few tubulin isotypes[25,26] resulting in comparatively homogeneous samples. We isolated tubulin from cell lysates by using a polymerization and depolymerization cycle to remove contaminations (Fig. 1a, b). Subsequently, we polymerized tubulin into MTs with high concentration of

1,4-piperazinediethanesulfonic acid (PIPES) to remove MAPs binding and stabilized MTs with Taxol[27]. As demonstrated by sodium dodecyl sulfate (SDS) polyacrylamide gel electrophoresis, Taxol-stabilized MTs showed a purity of 90% (Fig. 1b). To check functionality, we incubated our prepared MTs with a fluorescently tagged MTBD of MAP7 domain-containing protein 3 (MAP7D3, in the following abbreviated by MAP7), which served as a MT-binding probe, and found that the purified MTs were decorated by this protein fragment (Supplementary Fig. 1A, B, left). Notably, the binding of MAP7 induced some bundling of MTs (Supplementary Fig. 1A, B). However, the MT bundling does not affect the structure and the dynamics of tubulin C-terminal tails of each individual MT[28], and hence did not interfere with our ssNMR experiments presented below. We also investigated whether HeLa S3-derived tubulin can polymerize without Taxol and shows dynamic instability by preparing an additional sample without Taxol, leading to soluble tubulin after a subsequent depolymerization. The resulting tubulin was used in an in vitro reconstitution assay where MTs are polymerized from stable MT seeds with GFP-tagged end binding protein 3 (EB3) as a marker for growing ends. The tubulin we prepared polymerized into MTs and showed phases of growth and shortening (Fig. 1c and Supplementary Fig. 1C, Movie 1), demonstrating that the purified tubulin remains polymerization-competent using our protocol. Taken together, this protocol allowed us to purify functional tubulin from mammalian cells that could form MTs, thus providing the basis for our NMR experiments described below.

To obtain isotope-labeled MTs for ssNMR studies, we grew HeLa S3 cells in 2L [$^{13}$C, $^{15}$N] labeled medium and obtained isotope-labeled, Taxol-MTs (in the following abbreviated by MTs) using the procedure described above. The sample was then packed into a 1.3 mm MAS NMR rotor and subjected to extended measurement periods using fast MAS rates. To ensure sample integrity under such conditions, we used transmission EM and TIRF microscopy that confirmed that our MT samples remained intact (Fig. 1d and Supplementary Fig. 1B).

**MT samples as seen by $^{13}$C and $^{31}$P ssNMR spectroscopy.** To evaluate the sample quality, we first recorded a 2D $^{13}$C–$^{13}$C dipolar-based correlation spectrum using radio frequency-driven recoupling (RFDR)[29] recoupling and compared our data to chemical-shift predictions for MTs (PDB ID: 5SYF) using FANDAS 2.0[30]. Overall, the spectrum agreed with in silico estimates (Fig. 2a, left). For example, in Cα–Cβ regions of serine, threonine and alanine residues, signals were observed in qualitative agreement with predictions for α-helical and β-strand conformations (Fig. 2a, right). Interestingly, alanine signals that were not observed (Fig. 2a, top right) resided in random-coil conformation and related to residues in tubulin loops. The latter observation would be consistent with the occurrence of loop motion as discussed elsewhere[7] at the experimental temperature (~298 K) in our dipolar-based experiments. While residue-specific analysis of such effects will require the extension to 3D ssNMR spectroscopy, possibly in combination with amino acid specific labeling[11], the spectral resolution observed in Fig. 2 for such a large complex hence speaks in favor of a homogenous sample of polymerized MTs.

This notion was further confirmed by $^{31}$P NMR signals stemming from nucleotides binding to MTs. In general, α/β-tubulin dimers contain two nucleotide-binding sites. The GTP molecule bound to α-tubulin is non-hydrolysable, whereas, the nucleotide bound to β-tubulin is exchangeable, and GTP is hydrolyzed to GDP upon MT polymerization[31] (Fig. 2b, left). Indeed, we observed ssNMR signals from both GTP and GDP in

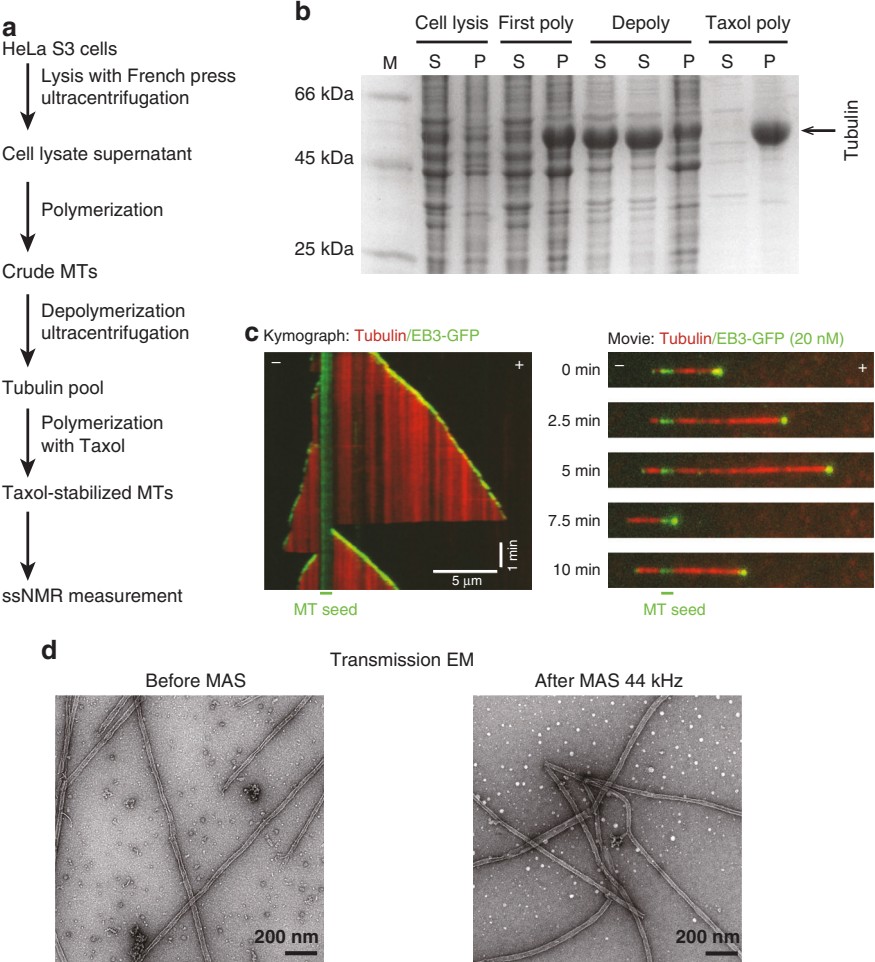

**Fig. 1 MT preparation from HeLa S3 cells and sample characterization. a** Schematic overview of the sample preparation. **b** SDS-PAGE analysis from each purification step. M: protein marker; S: supernatant; P: pellet; first poly: first polymerization; depoly: depolymerization; Taxol poly: polymerization with Taxol. The uncropped gel is provided as a Source Data file. **c** Functional characterization of purified HeLa S3 tubulin. Kymograph (left) and movie stills (right) of an in vitro polymerized MT imaged for 10 min on a TIRF microscope correspond to Fig. S1C. Time points and MT plus (+) and minus (−) ends are indicated. **d** Characterization of morphology of HeLa MTs before and after MAS NMR experiments by EM.

1D $^{31}$P CPMAS spectrum (Fig. 2b, right) that were well resolved compared to previous results on lyophilized MTs[32]. The observed spectral resolution allowed us to distinguish NMR signals from both molecules (Supplementary Table 1)[32,33]. Integrated peak intensities (Supplementary Table 1) correlated well with the notion that the molar ratio of GDP:GTP was 1:1, which is in agreement with previous studies[4,31] further supporting the idea that our ssNMR preparations were well folded and functional. Interestingly, we also observed an additional broad peak at −2.7 ppm, which may result from co-purified lipids[34].

**Probing tubulin tails by ssNMR**. To probe mobile MT protein segments including the C-terminal tubulin tails (Fig. 3a), we carried out a series of scalar-based (J-based) ssNMR experiments[35] that previously revealed flexible parts within large biomolecules[36–39]. As discussed elsewhere[36,40], significant molecular motions with correlation times in the range of $10^{-9}$ to $10^{-7}$ s must be present to generate NMR signals in such ssNMR correlation experiments. Slower motions (in the range of milliseconds or slower) will, on the other hand, lead to signal loss (vide infra). In the current context, a 2D $^{15}$N-HSQC spectrum contained several signals with a limited spectral dispersion (Fig. 3b), in line with the presence of a mobile unstructured

protein region. Using additional 2D $^{13}$C-HSQC (Fig. 3c, d) $^{15}$N-edited $^{1}$H–$^{1}$H spin diffusion spectra (Fig. 3d), we could confirm that correlations observed in the $^{15}$N-HSQC spectrum are mainly due to glycine, glutamate, and alanine, which are the most abundant residues in the tubulin tails (Fig. 3a). Moreover, the corresponding resonance frequencies seen in the 2D $^{13}$C-HSQC reflected random-coil chemical shifts of these amino acids. Note that the latter spectrum contained additional signals from mobile side chains as well as correlations stemming from residual lipids (Fig. 2b) and buffer.

We could further identify correlations stemming from glutamate and alanine residues by tracing HN to Hβ or Hγ correlations. Interestingly, one glutamate NH signal showed correlations to the Hα of glycine, indicating that the corresponding glutamate residues are next to glycine residues (Fig. 3d, second strip). Such correlations are only expected for the α-tubulin C-terminal tails (Fig. 3a). On the other hand, the other glutamate peak showed no contacts to glycine Hα (Fig. 3d, third strip), suggesting that these correlations most likely reflect glutamate residues of the β-tubulin C-terminal tail (Fig. 3a). In addition, we detected two glycine signals with an approximately 3:1 intensity ratio in line with three glycine residues in EGE motifs of the α-tubulin tail and one glycine representing the FGE motif of the β-tubulin. These tentative assignments were in good

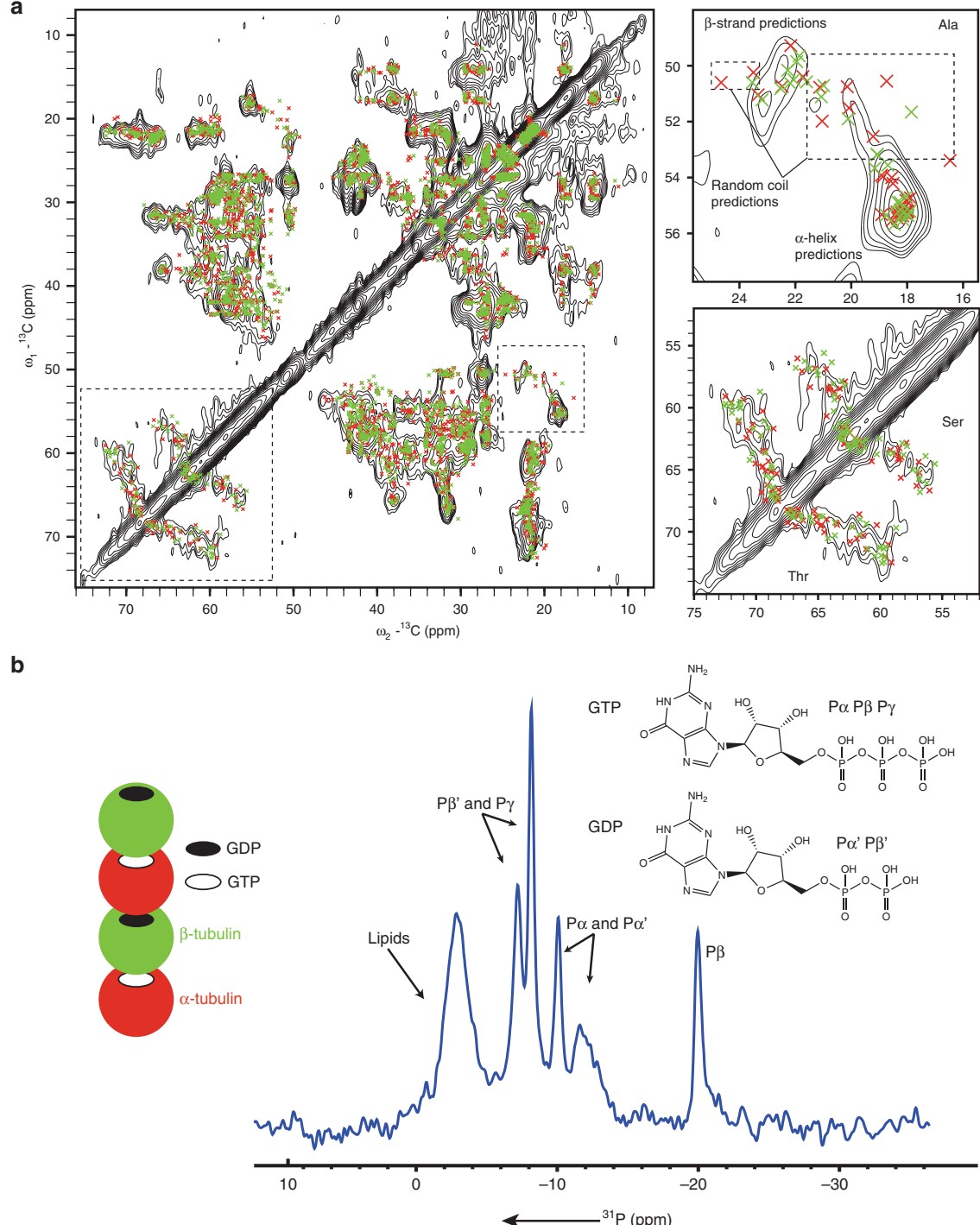

**Fig. 2 SsNMR spectra of [¹³C, ¹⁵N]-labeled MTs. a** The 2D ¹³C–¹³C radio frequency-driven recoupling (RFDR) experiment was performed at 270 K set temperature with a MAS rate of 44 kHz with zoom in's on serine, threonine and alanine Cα–Cβ regions. Crosses indicate the chemical shift predictions for MTs from SHIFTX2[57] and FANDAS 2.0[30] based on the EM structure (PDB 5SYF). Red and green crosses represent predictions for α-tubulin and β-tubulin, respectively. **b** Schematic representation of the GTP/GDP binding to tubulin in MTs (left), and 1D H–³¹P CP experiment at ambient temperature with MAS rate of 11 kHz allows to observe GTP and GDP bound to MTs.

agreement with previous solution-state NMR studies[41] of a peptide that contained C-terminal residues of human α-tubulin and enabled us to tentatively assign the most C-terminal residue of α-tubulin, i.e., Tyr451 (Fig. 3a). Moreover, we could identify one alanine correlation which most likely reflects the last residue of the β-tubulin based on its high ¹⁵N chemical-shift value, and the correlations from its amide proton to the Hβ of Ala and Glu.

Lastly, two correlations which may at least in part stem from the N-terminal residues (Fig. 3a) remained unassigned due to lack of cross peaks in the 3D spectrum (Fig. 3b, peaks indicated by asterisks). The corresponding resonance frequencies would speak against posttranslational modifications (PTMs) seen in previous solution-state NMR on *Tetrahymena thermophila* tubulin[42]. Indeed, earlier work showed that HeLa tubulin mostly lacks

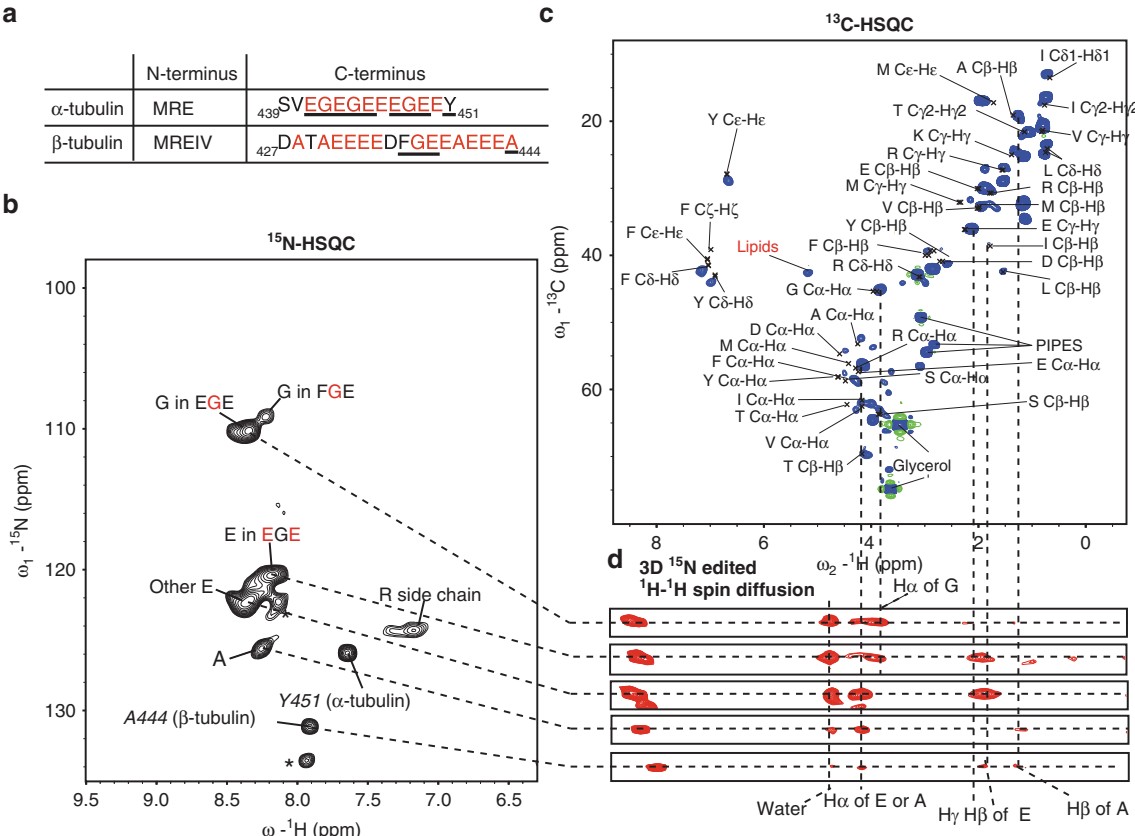

**Fig. 3 Flexible tubulin C-terminal investigated by J-based ssNMR. a** The sequences of N- and C-termini of tubulin. Underlined residues showed sequential correlations in our ssNMR data sets. The protein sequences of α1 and β1 tubulin isotypes were used for analysis based on previous results[26]. **b** 2D ${}^{15}$N-HSQC of [${}^{13}$C, ${}^{15}$N]-labeled MTs showed the flexible tubulin tails. **c** 2D ${}^{13}$C-HSQC of [${}^{13}$C, ${}^{15}$N]-labeled MTs showing the residues present in the tubulin tails. Due to the high flexibility, the signals from side chains of leucine, lysine as well as buffer compounds and lipids were also observed. **d** Strips of a 3D ${}^{15}$N-edited ${}^{1}$H–${}^{1}$H spin diffusion experiment showing the connections between ${}^{13}$C-HSQC and ${}^{15}$N-HSQC.

PTMs on tubulin tails[43]. Taken together, our results suggested that both α- and β-tubulin tails are disordered and highly mobile on a nanosecond time scale.

**Tubulin tail dynamics are modulated by the CKK domain.** In our previous work we could decipher molecular properties that enable CKK domains of the CAMSAP protein family to preferentially associate with the transition zone between curved protofilaments and the regular MT lattice at MT minus ends[15,19]. We found that the CKK domain that consists of folded protein core and disordered termini associates with the groove between two laterally attached tubulin dimers and discovered that CKKs can subtly differentiate specific tubulin conformations to enable MT minus-end recognition[19]. However, these studies did not reveal how tubulin tails, which are important for CKK binding[15] are involved in these events. Using our ssNMR protocol, we in the following investigated the influence of CKK binding upon the C-terminal tubulin tails. For this purpose, we purified the unlabeled CAMSAP1 CKK N1492A mutant that exhibits reduced minus-end selectivity and binds with higher affinity to MTs compared to wild-type CKK[15,19]. We then added unlabeled CKK to [${}^{13}$C, ${}^{15}$N]-labeled MTs and repeated our ${}^{15}$N-HSQC experiments. Compared to free [${}^{13}$C, ${}^{15}$N]-labeled MTs, we observed strong changes in the 2D correlation spectrum (Fig. 4a) that relate to arginine residues of MTs as well as to the C-terminal residues of α-tubulin and of β-tubulin. In the following, we discuss our ssNMR spectral findings and their relationship to previous results for each of these three aspects.

Firstly, signals for the side chains of arginine residues were strongly reduced compared to free MTs. This observation confirmed our previous findings that arginine residues of both α- and β-tubulin tubulin are located in the binding region of the CKK domain[15] (Fig. 4b). Furthermore, our recent studies also showed that the binding of CKK induces protofilament skew leading to a reduced lateral space between tubulin dimers[15,19]. Arginine residues located in regions distant from the CKK binding epitope but involved in these lateral interactions between tubulin may hence also become less dynamic[44].

Secondly, several correlations including the last C-terminal residue of β-tubulin, A444, disappeared after CKK was bound to MTs and correlations appeared in the alanine, glycine and tyrosine regions (Fig. 4a). These spectral changes would be consistent with conformational exchange on the fast time scale[9] upon CKK binding. Interestingly, glycines in the FGE segment and alanine residues are only found in the C-terminal tail of β-tubulin suggesting that the β-tubulin tail is involved in this dynamic interaction. Notably, the C-termini of both β-tubulin (17 residues, Fig. 4c) and of the CKK domain (13 residues, Fig. 4c) were not observed in the previous cryo-EM structure[15]. Most likely, the CKK C-terminus hence retains conformational variability upon binding to MTs. Assuming fully extended backbone conformations, both CKK and tubulin C-terminal tails could sample a distance range of at least 60 and 45 Å[45], respectively, which would readily allow for such dynamic interactions between both mobile tails as indicated in Fig. 4d. Notably, previous EM and ssNMR experiments suggested that the N-terminus of CKK (9 residues) could interact with the C-

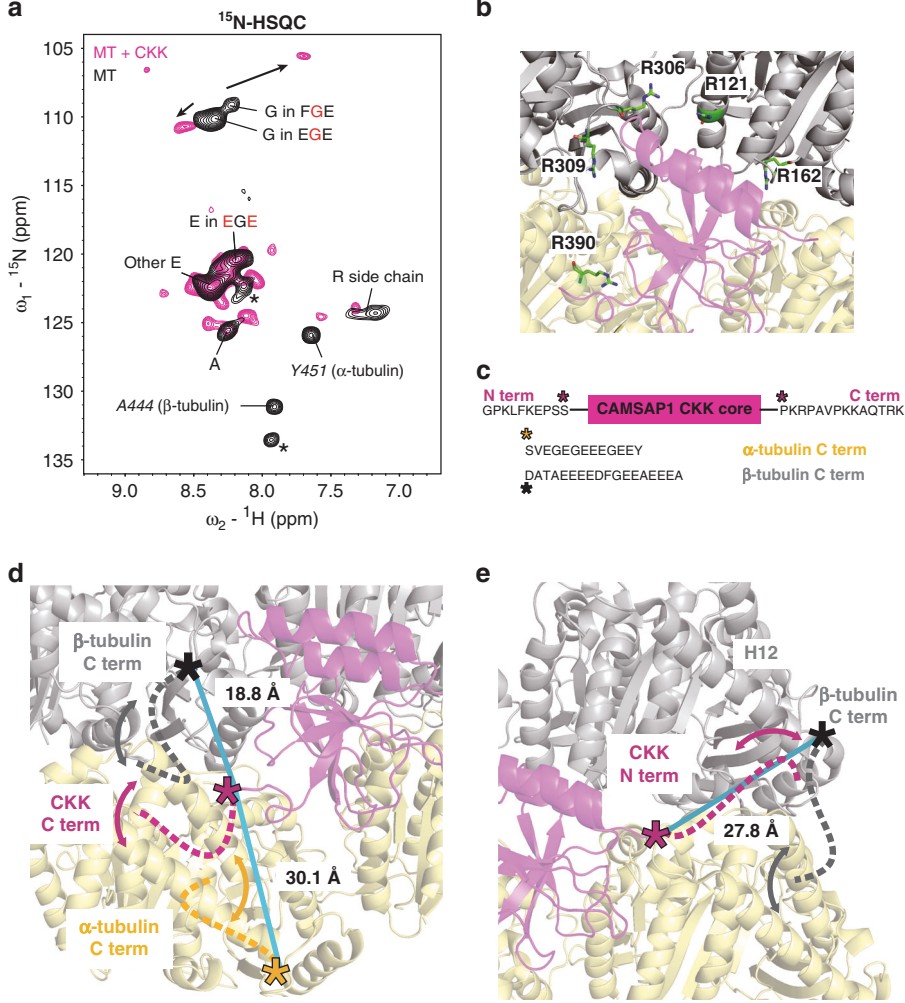

**Fig. 4 Tubulin C-terminal tail dynamics are modulated by the CKK domain. a** Comparison of 2D $^{15}$N-HSQC spectra obtained from [$^{13}$C, $^{15}$N]-labeled MTs (black) and unlabeled CAMSAP1 CKK N1492A in complex with [$^{13}$C, $^{15}$N]-labeled MTs (magenta). **b** Zoom-in on arginine residues found in the CKK-MT cryo-EM structure (PDB 5M5C), ref. [15]) that are located on α-tubulin (yellow) and β-tubulin (gray) and within 6 Å to CKK domain (magenta). **c** Top: amino-acid sequences of CKK termini with last residues observed in the cryo-EM structure (PDB 5M5C) indicated by stars. Bottom: C-terminal tails of α-tubulin and β-tubulin with last residues detected the cryo-EM structure (PDB 5M5C) are indicated by yellow and black stars, respectively. **d** Dynamic (indicated by double-headed arrows) C-terminal tubulin (dashed black and yellow lines) and CKK (magenta) tails. Blue lines measure the distance between the last residues observed in the cryo-EM structure of both tubulin tails (indicated by black and yellow stars) to the last CKK C-terminal residue observed in cryo-EM (magenta star). **e** Dynamic (indicated by double-headed arrows) CKK N-terminus (dashed magenta line) and β-tubulin C-terminus (dashed black line). The blue line indicates the distance between the last residue of β-tubulin and the first residue of CKK detected in the cryo-EM structure.

terminal part of β-tubulin (Fig. 4c, e), which would be in line with our NMR data.

Thirdly, we also observed changes in ssNMR signal frequencies and intensities after CKK binding for glycine in EGE segments and residue Y451, respectively, that are both found in the C-terminal tail of α-tubulin (Fig. 4a). As shown in Fig. 4c, d, the C-terminal tail of α-tubulin with its 12 residues could hence also dynamically interact with the C-terminus of CKK.

Taken together, our results strongly suggest that that the C-terminal tubulin tails dynamically interact with CKK domain on the nano- to microsecond time scale. Combination of these results with our previous EM and ssNMR data further refines this notion, suggesting that the CKK C-terminus interacts with both α- and β-tubulin while the CKK N-terminus engages in fast interactions with the C-terminus of β-tubulin. Future ssNMR experiments may allow us to further dissect the details of these dynamic interactions, including the more precise determination of motional time scales in a residue-specific manner.

**MAP7-binding reduces MT tail dynamics**. The MAP7 family of proteins plays an important role in regulating kinesin-based intracellular transport. However, the structural details of MAP7–MT interaction are currently unknown[20–22]. A secondary-structure prediction of MAP7 on the basis of its amino acid sequence as well as preliminary NMR experiments on free MAP7 in solution suggest that the MAP7 domain adopts a highly α-helical structure separated by a short loop comprising R144 to T146 and flanked by unstructured termini (Fig. 5a).

To obtain insight into the interaction of MT tails with bound MAP7, we repeated our J-based experiments on [$^{13}$C, $^{15}$N]-labeled MTs after adding unlabeled MAP7 in a 2:1 ratio ensuring full decoration of MTs (see Methods). Interestingly and unlike in the case of binding of the CKK domain to MTs, we did not detect any frequency changes in the 2D $^{15}$N-HSQC ssNMR correlation spectrum (Fig. 5b, cyan) compared to the case of free MTs. Instead, signals from both C-terminal tubulin tails, including the identified EGE and FGE segments as well as other E residues were

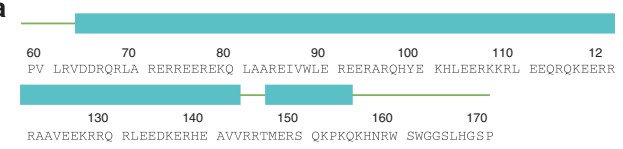

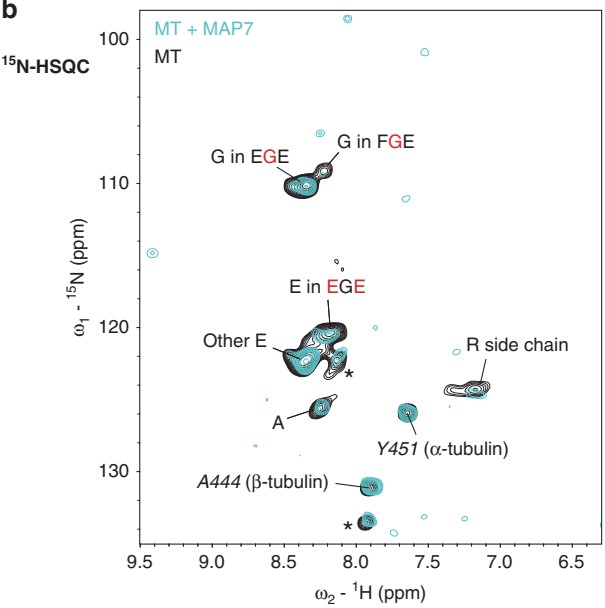

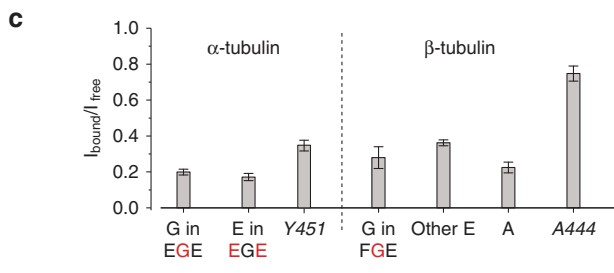

**Fig. 5 Tubulin C-terminal tails interact with MAP7. a** Secondary structure prediction of MAP7-MTBD by JPred[58]. **b** Spectral comparison of 2D $^{15}$N-HSQC spectra obtained from [$^{13}$C, $^{15}$N]-labeled MTs (black) and unlabeled MAP7-MTBD in complex with [$^{13}$C, $^{15}$N]-labeled MTs (cyan). **c** Signal intensity ratios of resolved peaks in $^{15}$N-HSQC between MAP7-MTBD bound MTs and free MTs. Error bars were calculated based on the NMR signal to noise ratio of each peak in the two spectra. Signal intensity ratios are provided as a Source Data file.

strongly attenuated (Fig. 5c and Supplementary Fig. 2) compared to the most C-terminal residues of α- and β-tubulin, Y451 and A444, respectively. These observations can be explained by a slow exchange process (millisecond time scale or longer) between freely mobile α- and β-tubulin tail conformations (visible in the spectra) and C-terminal tails directly bound to MAP7. Note that the latter states would be invisible in our J-based experiments. In a simple two-state model, the relative amount of bound populations would be given by $P_{bound} = 1 - I_{bound}/I_{free}$ (see, e.g., refs. [9,46]) seen in Fig. 5c leading to values of up to a 80% bound tail residues around the EGE and FGE terminal segments in α and β tubulin tails, respectively, and decreasing to around to a 65% and 25% bound population for the most C-terminal residues of α-tubulin (Tyr$_{451}$) and of β-tubulin (Ala$_{444}$), respectively. Unlike the case of CKK where dynamic tubulin C termini retain fast dynamics, these observations hence suggested that the MT termini directly bind to MAP7 at a time scale of milliseconds or longer. At the same time the formation of the MT–MAP7

complex also leads to the reduction of Arg side-chain dynamics which are apparent from our experimental data (Fig. 4b) and reminiscent of our findings in the case of CKK binding.

## Discussion

Protein dynamics are critically involved in MT organization including the stochastic switching between growing and shrinking states as well as in the regulation of MTs by MAPs and other cellular factors. NMR carries the potential to directly probe such processes provided that functional and NMR-active MTs can be obtained. We have shown that high-resolution ssNMR spectra can be recorded on [$^{13}$C, $^{15}$N] labeled MTs obtained from human cells. The MTs were polymerized from tubulin in functional form[24], as confirmed by in vitro MT dynamics assays. In addition, our protocol allows obtaining purified, functional tubulin if Taxol is not included in the second polymerization (see Fig. 1c). Such preparations allow for NMR studies of human tubulin in solution (or after depolymerization), e.g., to study MT nucleation and its regulation by different MAPs and potentially related phase transitions[47]. The narrow $^{31}$P NMR lines indicated homogenous binding of nucleotides on tubulin subunits that resulted in correctly polymerized MTs. This notion was further confirmed by 2D dipolar-based correlation experiments that are in qualitative agreement with a MT lattice as seen in cryo-EM structures. The absence of loop signals in our 2D ssNMR dipolar based data possibly points towards enhanced dynamics in protein loop regions as expected for tubulin domains, which are critical for MT dynamic instability or involved in PTMs such as the α-tubulin loop of residues P37–D47 (also known as αK40 loop[7]). Further ssNMR studies, possibly using three- or higher-dimensional ssNMR experiments may, in combination with tailored labeling, reveal the exact details of these motions.

Complementary to these experiments, proton-detected scalar-based 2D and 3D ssNMR experiments allowed us to directly study the disordered, flexible tubulin tails that are critically involved in MAP binding[41] and which are invisible in all available high-resolution structures. In the current context, we tracked the effect of binding of the CKK domain of CAMSAP and of MAP7 for the dynamics of α- and β-tubulin C-terminal tails. Our data suggest that MAP-tubulin tail interactions can take place over a remarkably wide time scale. In the case of CKK, the dynamics of both α- and β-tubulin C-terminal tails are rapidly (in the nano to microsecond scale) modulated by the CKK domain, confirming our previous finding that tubulin C-terminal tails are important for CKK binding. Combination of these experiments with results of our earlier cryo-EM and ssNMR studies[15,19] provided further insight regarding the details of these dynamic interactions and suggest that the disordered termini of CKK are involved in these interactions. On the other hand, MAP7 binding to MTs is characterized by tubulin C-terminal tails that exchange between bound (major state) and free (minor state) conformations on a much slower (millisecond or longer) time scale.

The remarkable difference between CKK and MAP7 binding to the tubulin tails may be related in part to the structural and dynamical properties of the MAP itself. For example, the mobile CKK termini comprise several positively charged amino acids (Fig. 4c), whereas, in the case of MAP7, such residues are mostly found in the putatively structured core of MAP7 (Fig. 5a). In addition, the molecular details of the MAP binding epitope on MTs may play a role in defining tail interactions. As we have shown earlier[15,19], CKK binding takes place between protofilaments. On the other hand, MAP7 is known to compete for binding to MTs[20] with the protein tau and hence is likely to be associated with the protofilament crest as seen for tau[6]. Further ssNMR studies may help to dissect these dependencies, including

the exact CKK and MAP7 regions interacting with the C-terminal tails. Such studies may involve isotope labeled MAPs or mixed labeled[48] MAP-MT preparations. Moreover, the presented approach could be readily extended to study the interactions of longer variants of MAPs that, in addition to the minimal MT binding region, comprise intrinsically disordered protein domains as well as the Kinesin-1 binding domain[22]. As we have shown recently for the case of mRNA processing bodies[39], such studies not only could target bona fide MTs but they could also reveal dynamic interactions leading to the compartmentalization of the MT lattice by condensation of tau or other MAPs[47,49,50]. Combination of such NMR experiments with data obtained from cryo-EM and fluorescence microscopy experiments could lead to a comprehensive view on the structural and dynamical aspects that define the interaction between MT and MAPs or other cellular players that regulate MT function.

## Methods

**Preparation of MTs for ssNMR experiments and in vitro assays.** The HeLa S3 cell line (ATCC® CCL2.2™) was used for culturing. [$^{13}$C, $^{15}$N] labeled DMEM medium was prepared in the same way as described in a previous study[51] except that we used 3.5 g/L glucose in order to maintain cell viability in suspension culture. The cells were first cultured in the labeled medium on two 150 mm cell culture dishes, and then transferred into 12 dishes with the same medium. When the culture reached a confluence of ~80% on the plates, cells were trypsinized and transferred into 2 L labeled medium and cultured in 7 1 L Erlenmeyer shaker flasks (Corning) until the cell density reached ~1.2–1.5 × 10$^6$/mL. Cells were then harvested by centrifugation at 500 × g for 20 min at 4 °C. The cell pellet was collected and resuspended in phosphate-buffered saline and centrifuged again at 500 × g for 15 min at 4 °C and used for MT preparation.

The purification of Taxol-stabilized MTs was performed based on the published protocols[52,53] with minor modifications. Harvested cells were first resuspended with 1 g cell/mL lysis buffer (80 mM PIPES pH 6.8, 1 mM EGTA, 1 mM MgCl$_2$, 1 mM β-mercaptoethanol, 1 mM PMSF and protease inhibitors (Roche) and lysed on ice by passing through a French Press homogenizer 3 times at 1000 psi. Subsequently, lysed cells were spun down at 120,000 × g at 4 °C for 30 min and the supernatant was collected. The supernatant was centrifuged again at 5000 × g at 4 °C for 15 min to remove the remaining cell debris and then a half volume of glycerol and 1 mM GTP were added and mixed well. MT polymerization was carried out by incubating the mixture at 30 °C for 30 min. Subsequently, the crude MT pellet was spun down at 150,000 × g (Type 70.1 Ti, Beckman Coulter) at 30 °C for 30 min and placed on ice. The pellet was then resuspended in BRB80 buffer (80 mM PIPES pH 6.8, 1 mM EGTA, 1 mM MgCl$_2$) supplemented with protease inhibitors and kept on ice for 30 min to allow for MT depolymerization. For a more efficient depolymerization, the solution was resuspended frequently and then centrifuged at 150,000 × g (Type 70.1 Ti, Beckman Coulter) at 4 °C for 30 min, and the supernatant was collected. An equal volume of high concentration PIPES buffer (1 M PIPES pH 6.8, 10 mM MgCl$_2$, 20 mM EGTA), together with an equal volume of glycerol and 1 mM GTP were then added to the supernatant and mixed well. MT polymerization was again performed at 30 °C for 30 min. Subsequently, 20 μM Taxol (Paclitaxel, Sigma) was added to the reaction and incubated for 20 min to generate Taxol-stabilized MTs. Taxol-stabilized MTs were then spun down at 150,000 × g (TLA-55, Beckman Coulter) at 30 °C for 30 min washed with BRB80 containing 20 μM Taxol and protease inhibitors.

For all in-vitro assays, Taxol-stabilized MTs were prepared in the same way as described above. Regarding the sample used in the MT dynamics assay, the preparation of tubulin was similar except that Paclitaxel was not included during the second MT polymerization. After the second MT polymerization, MTs were spun down and resuspended in ice cold BRB80 supplemented with protease inhibitors. The second depolymerization was done on ice for 30 min and then centrifuged at 150,000 × g at 4 °C for 30 min. Purified tubulin was aliquoted and snap-frozen in liquid nitrogen, stored at −80 °C until use.

**In vitro assays.** MT seeds were prepared by incubating 20 μM porcine tubulin mix containing 70% unlabeled, 18% biotin–tubulin and 12% HiLyte488-tubulin with 1 mM guanylyl-(α,β)-methylenediphosphonate (GMPCPP) at 37 °C for 30 min. Polymerized MTs were separated from the mix by centrifugation in an Airfuge for 5 min. MTs were subjected to one round of depolymerization and polymerization in 1 mM GMPCPP, and the final MT seeds were stored in MRB80 buffer (80 mM K-PIPES pH 6.8, 1 mM EGTA, 4 mM MgCl$_2$) containing 10% glycerol.

In vitro reconstitution assays were performed in flow chambers. Flow chambers were assembled by sticking plasma-cleaned glass coverslips on microscopic slides with double-sided tape. Assays with Taxol-stabilized HeLa MTs were performed in MRB80 prewashed chambers. The in vitro reaction mixture consisted of 15 μM HeLa MTs (stabilized by 20 μM Taxol) that were either before or after MAS at 44 kHz for 24 h, 150 nM MAP7D3-MTBD, 50 mM KCl, 0.1% Methylcellulose, 0.5 mg/

ml κ-casein, 1 mM GTP, oxygen scavenging system (20 mM glucose, 200 μg/ml catalase, 400 μg/ml glucose-oxidase, 4 mM DTT). After centrifugation in an Airfuge for 5 min at 119,000 × g, the reaction mixture was added to the flow chamber and sealed with vacuum grease.

Dynamic MT assays were performed in flow chambers that were functionalized by sequential incubation with 0.2 mg/ml PLL-PEG-biotin (Surface Solutions, Switzerland) and 1 mg/ml NeutrAvidin in MRB80 buffer for 5 min. MT seeds were attached to the biotin-NeutrAvidin links and incubated with 1 mg/ml κ-casein. The in vitro reaction mixture consisted of 15 μM HeLa tubulin (quantified using bovine serum albumin standard), 20 nM GFP-EB3 and 0.5 μM rhodamine-labeled porcine brain tubulin, 50 mM KCl, 0.1% methylcellulose, 0.5 mg/ml κ-casein, 1 mM GTP, oxygen scavenging system (20 mM glucose, 200 μg/ml catalase, 400 μg/ml glucose-oxidase, 4 mM DTT). After centrifugation in an Airfuge for 5 min at 119,000 g, the reaction mixture was added to the flow chamber containing the HiLyte-488 MT seeds and sealed with vacuum grease. HiLyte488- and rhodamine-tubulin were purchased from Cytoskeleton Inc.

All in vitro experiments were conducted at 30 °C. Data were collected using total internal reflection fluorescence (TIRF) microscopy on an inverted research microscope Nikon Eclipse Ti-E (Nikon) with the perfect focus system (Nikon), equipped with Nikon CFI Apo TIRF 100× 1.49 N.A. oil objective (Nikon, Tokyo, Japan), Photometrics Evolve 512 EMCCD (Roper Scientific) and Photometrics CoolSNAP HQ2 CCD (Roper Scientific) and controlled with MetaMorph 7.7 software (Molecular Devices, CA). The microscope was equipped with TIRF-E motorized TIRF illuminator modified by Roper Scientific France/PICT-IBiSA, Institut Curie. For excitation lasers we used 491 nm 100 mW Stradus (Vortran), 561 nm 100 mW Jive (Cobolt) and 642 nm 110 mW Stradus (Vortran). We used an ET-GFP 49002 filter set (Chroma) for imaging of proteins tagged with GFP and an ET-mCherry 49008 filter set (Chroma) for imaging X-rhodamine labeled tubulin amd mCherry-labeled MTBD of MAP7D3. For simultaneous imaging of green and red fluorescence we used an Evolve512 EMCCD camera (Photometrics), ET-GFP/mCherry filter cube (59022, Chroma) together with an Optosplit III beamsplitter (Cairn Research Ltd.) equipped with double emission filter cube configured with ET525/50m, ET630/75m, and T585lprx (Chroma). To keep in vitro samples at 30 °C, we used a stage top incubator (model INUBG2E-ZILCS; Tokai Hit). Images were processed using ImageJ. All images were modified by linear adjustments of brightness and contrast. Kymographs were generated using ImageJ plugin KymoResliceWide v.0.4. https://github.com/ekatrukha/KymoResliceWide; copy archived at https://github.com/elifesciences-publications/KymoResliceWide).

**Transmission EM.** Taxol-stabilized MTs were prepared with and without MAS spinning as described above. The protein samples were negatively stained with 2% uranyl acetate on glow-discharged carbon coated copper grids. Images were recorded on a Tecnai 20 electron microscope with a LaB6 filament, operating at 200 kV with a BM Eagle 4 K CCD camera (ThermoFisher, Eindhoven, The Netherlands). Images were acquired with a defocus of approximately 5 μm The magnification used at 19,000 × g resulting in an effective pixel size of 1.14 nm on the specimen level.

**Preparation of CAMSAP1 CKK N1492A in complex with MTs.** [$^{13}$C, $^{15}$N] labeled MTs were prepared as described above. The purification of CKK domain and preparation of CKK–MT complexes were performed based on the method published in ref. [15]. CKK was first purified by a ÄKTA pure system with a POROS™ MC column that was saturated with Ni$^{2+}$. The column was first equilibrated using washing buffer (50 mM phosphate buffer, pH 8, 500 mM NaCl, 1 mM β-mercaptoethanol and 20 mM imidazole). The protein was eluted with the same buffer but containing 400 mM imidazole. Subsequently, the protein was loaded onto a SEC HiLoad Superdex 75 26/60 column (GE Healthcare) equilibrated with 40 mM phosphate buffer, pH 7, supplemented with 150 mM NaCl and 1 mM DTT. The purified protein was then concentrated and used for ssNMR sample preparation. CKK N1492A was then added to a final concentration of 65.3 μM (4:1 CKK/tubulin) and incubated with labeled MTs at 37 °C for 30 min. The pellet was centrifuged down at 180,000 × g (Beckman TLA-55 rotor) at 30 °C for 30 min and washed with BRB80, without disturbing the pellet. The pellet was then transferred and packed into a 1.3 mm rotor.

**Preparation of MAP7-MTBD and of MAP7-MTBD- MT complexes.** The cDNA of human MAP7-MTBD (residues 59–170) was cloned into the pLICHIS vectors with the gene encoding a N-terminal His tag-Maltose-binding protein (MBP)-thrombin cleavage site by using ligation independent cloning[54].Transformation was done with *Escherichia coli* Rosetta 2 cells and grown in 1 L unlabeled M9 medium. Induction of the protein was done when OD$_{600}$ reached 0.6 with 0.3 mM IPTG at 20 °C for 16 h. The cultures were then centrifuged with 4000 × g at 4 °C for 20 min to harvest cells. Cell pellets were washed with 50 mM sodium phosphate buffer, pH 8, 500 mM NaCl, 1 mM β-mercaptoethanol and 20 mM imidazole and store at −80 °C.

For protein purification, cell lysis was done by sonication on ice and the cell lysate was collected by centrifugation of 40,000 × g at 4 °C for 30 min. Subsequently, the proteins were purified by a ÄKTA pure system with a POROS™ MC column that was saturated with Ni$^{2+}$. The column was first equilibrated with

the same buffer as above mentioned. The cell lysate was then loaded onto the column and the column was washed with 20 column volumes with the same buffer. Proteins were eluted with 50 mM sodium phosphate buffer, pH 8, 150 mM NaCl, 1 mM β-mercaptoethanol and 400 mM imidazole supplemented with protease inhibitors. The protein was concentrated and then diluted with 40 mM sodium phosphate buffer, pH 6.5 to reach a final concentration of imidazole of 20–30 mM. Subsequently, a cation exchange purification was performed with the Hitrap HP SP chromatography column (GE Healthcare Life Sciences). The column was first equilibrated with buffer A (40 mM sodium phosphate buffer, pH 6.5). The sample was then loaded onto the column and washed 5 column volumes with buffer A. A gradient elution was used by combining buffer A and buffer B (40 mM sodium phosphate buffer, 1 M NaCl, pH 6.5) to elute the protein. The protein was then concentrated and stored at 4 °C. Finally, thrombin was used to cleave the MBP fusion and then supplemented with protease inhibitors.

A 10 mg/ml porcine brain tubulin was diluted in BRB80 buffer (80 mM K-PIPES, pH 6.8, 1 mM EGTA, 1 mM $MgCl_2$) to 2 mg/ml. After the addition of 1 mM GTP, the sample was incubated on ice for 5 min. MT polymerization was performed at 30 °C for 20 min. Subsequently, 20 μM taxol was added and incubated for 15 min at 30 °C. MT pelleting assays in the presence of MAP7-MTBD were performed by mixing with different molar ratios of MAP7-MTBD:MTs. As control treatments, taxol-stabilized MTs or MAP7-MTBD was applied alone. All the samples were then centrifuged at 180,000 × g for 30 min at 30 °C, an aliquot was taken from the supernatant. After removal of the supernatant, the pellet was resuspended in SDS sample buffer. Samples of supernatant and pellet fractions were loaded and analyzed on Coomassie-stained 12.5% SDS gels.

Before sample preparation for ssNMR, the purified unlabeled MBP-MAP7-MTBD was first incubated with 5 U thrombin at 4 °C for 16 h to cleave the fused MBP protein. Protease inhibitors were then added to the proteins. To prepare MAP7-MTBD-MT complexes, 2 mg of purified [$^{13}$C, $^{15}$N] labeled MTs were prepared and resuspended in warm BRB80 buffer with 20 μM taxol. MAP7-MTBD was then added to molar ratio of 2:1 for MAP7-MTBD:MT and incubated at 30 °C for 30 min. The pellet was centrifuged down at 180,000 × g (Beckman TLA-55 rotor) at 30 °C for 30 min and washed with warm BRB80 buffer with 20 μM taxol and protease inhibitors, without disturbing the pellet. The pellet was then transferred and packed into a 1.3 mm rotor.

**ssNMR experiments**. NMR experiments of [$^{13}$C, $^{15}$N] labeled MTs (free MTs, CKK-bound MTs and MAP7-bound MTs) were performed on a standard-bore 700 MHz, a wide-bore 500 MHz (Bruker Biospin). SsNMR experiments on 700 MHz included 2D $^{13}$C–$^{13}$C RFDR[29], $^1$H–$^{13}$C/$^{15}$N HSQC and 3D $^{15}$N-edited $^1$H–$^1$H spin diffusion experiments[35] (set temperature 270 K, MAS rate 44 kHz). The actual temperature during the ssNMR experiments for both apo as well as MAP-bound microtubular samples was 293 K. The $^{13}$C–$^{13}$C mixing was 3 ms in the RFDR experiment. The PISSARRO decoupling[55] scheme of 120 kHz was employed on the $^1$H channel during the RFDR mixing and the detection of $^{13}$C. The proton spin diffusion time was 200 ms. In all proton-detected experiments, the PISSARRO decoupling scheme was applied at 11 kHz on $^1$H, $^{13}$C and $^{15}$N channels. $^{31}$P NMR experiment was conducted on the 500 MHz spectrometer (set temperature 290 K, MAS rate 11 kHz). The CP MAS experiment was recorded using a 100–50% ramp on $^1$H-channel of 95.4 kHz, and 71.7 kHz on the $^{31}$P-channel, with 1.2 ms CP contact time. A SPINAL decoupling[56] of $^1$H during the acquisition of $^{31}$P was applied at 90 kHz. $^{31}$P chemical shifts were referenced externally using phosphate buffer, pH 7 and set the signal at 0 ppm.

**Reporting summary**. Further information on research design is available in the Nature Research Reporting Summary linked to this article.

## Data availability
Data supporting the findings of this manuscript are available from the corresponding authors upon reasonable request. A reporting summary for this Article is available as a Supplementary Information file. The source data underlying Figs. 1b and 5c and Supplementary Figs. 2 and 3 are provided in a Source Data file.

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

## Acknowledgements

We thank Gert Folkers for helpful discussions and Johan van der Zwan for technical support. This work was supported by the Dutch Science Foundation NWO: VENI grant 722.016.002 to S.X., ALW Open program grant 824.15.017 to A.A., NWO-Groot (no. 175.010.2009.002) and TOP-PUNT (no. 718.015.001) grant to M.B. and by uNMR-NL, the National Roadmap Large-Scale NMR Facility of the Netherlands (grant 184.032.207). C.J. is supported by the grants ANR-10-IDEX-0001-02 PSL, ANR-11-LBX-0038, FRM DEQ20170336756. S.B. was supported by the FRM grant FDT201805005465, and CEFIPRA 5703-1, and J.A.S. by the European Union's Horizon 2020 Marie Skłodowska-Curie grant agreement No. 675737, and the FRM grant FDT201904008210.

## Author contributions

Y.L., S.X., A.A., and M.B. designed the experiments, analyzed the data and wrote the paper. Y.L. and P.J.H. produced labeled microtubules with advise from J.A.S. and C.J. Y.L. and S.X. performed and analyzed the NMR experiments; P.J.H. performed the in vitro assays. L.v.B. and F.F. performed the transmission EM experiments. All authors critically reviewed the paper.

## Competing interests

The authors declare no competing interests.
