## [Peer Review File · Nature Communications]

Reviewers' Comments:

Reviewer #1:

Remarks to the Author:

The authors successfully generated isotope-labelled tubulin and assessed the binding of microtubules and MAP7 by solid state NMR. They managed to detect differences in the C-terminal portion of tubulin in the absence and presence of MAP7 using ssNMR, indicating an interaction of MAP7 through tubulin C-terminal tail.

It is nice to see that this type of experiment provides insights, and potentially applicable to assess binding of unstructured protein regions to interaction partners.

I am however concerned if there is enough novelty in this study and I am afraid that it would be better suited for a more specialized journal.

Specific concerns are as follows:

The authors can assign peaks for glutamate, glycine and alanine for the interaction at the C-terminal tail of tubulin. However it is not clear if this is really a generally applicable observation for other MAPs.

Can the authors do a control experiment to see the interaction to other proteins that are supposed to bind to the tubulin C-terminus, such as Tau and CAP-Gly domain-containing proteins?

Further it seems necessary to do a negative control experiment with proteins that do not primarily interact with the C-terminal tail. For example, kinesin, as the authors discussed in the manuscript.

The model is rather confusing and it does not add any new information learned from this study.

The problem is that the model is based on a previously published observation that MAP7 and Tau compete and not by the data from this study.

Minor:

In the discussion the authors mention the MAP7 binding competes with tau but not with kinesin-1. This should be more carefully mentioned, as the authors only used the MTBD of MAP7 and the entire MAP7 is much larger. Competitive binding does not only happen through the small binding sites for the tubulin tails but in a larger context such as electrostatic interactions and steric hindrance.

Which R and K residues in MAP7 are in close proximity to H12 helix?

It looks like the binding of MAP7 to microtubules causes bundling (S.fig 1A and B). The authors should comment if there is any effect in the ssNMR measurement.

Reviewer #2:

Remarks to the Author:

The authors report a solid-state NMR study on the interaction of microtubules (MTs) with MT-associated proteins (MAPs), which are regulated by GDP/GTP binding. MAPs are intrinsically disordered and seem to interact with the dynamic MT surface. The molecular details are unknown.

The authors have prepared ¹³C-¹⁵N-MT from human HELA cells. They are functional and well-folded, which is a remarkable achievement. Their protocol relies on the use of HELA S3 cells and the application of polymerization/depolymerization cycles. MT was stabilized by taxol.

Isotope labelling of MT enabled the recording of first ¹³C-¹³C MAS-NMR spectra. Microscopic controls before and after MAS sample spinning confirmed sample integrity.

A qualitative comparison of the dipolar ¹³C spectra with empirical chemical predictions revealed a

good match for the Ser/Thr Ca/Cb region and lack of residues in the Ala Ca/Cb area. The latter is explained by loop motions.

Furthermore, bound GDP and GTP could be detected by ³¹P-cross polarization. Here, well-resolved spectra could be obtained indicating good sample homogeneity.

Scalar-based experiments confirmed high mobility and disorder of residues with the alpha- and beta-tubulin tails.

Upon binding of MAP7, the intensity of some peaks is reduced, which supports that the corresponding residues are involved in binding.

Combining these observations with the MAP7 secondary structure and known biochemical observations regarding protein tau and kinesin-1 binding, a cartoon as derived in which the potential location of MAP7 is indicated.

Overall, this is a solid study, which reports important progress. It is still on a very qualitative / proof-of-concept stage but the possibility to produce such samples enables a range of novel experiments as the authors correctly point out in their outlook.

I only have a couple of questions/suggestions:

- The authors also describe the preparation of MT without taxol and refer to Fig. 1c, but it was not clear to me to which conclusion these data lead in the context of this paper. Additional explanations would be helpful.
- It appears that the lipid and P_alpha' peaks in Fig. 2b are broader than the other resonances. Did the authors record Hahn-echo experiments or spectra at different CP contact times to further characterize the dynamics of the bound species?
- The model in Fig. 4 is more a cartoon. Would it be possible, already based on these data, to derive a better model for example based on a docking/MD approach?

Reviewer #3:

Remarks to the Author:

The manuscript by Luo et al. reports the observation of the interaction of microtubules associated protein MAP7 with human microtubules using solid-state NMR spectroscopy.

There are already a number of studies of isotope labeled MAP proteins with microtubules, but the inverse experiment, taking isotope enriched tubulin could not be performed since the expression yield of labeled tubulin was not sufficient for NMR studies. The authors describe the application of a labeling procedure in HELA cells to tubulin that has been published earlier by them for EGFR. In fact they obtained natively ¹⁵N, ¹³C enriched folded tubulin. They show solid state NMR spectra of tubulin. This is an important step for further solid state studies of microtubules. By scalar-based (J-based) ssNMR experiments they can identify some signals from mobile segments of tubulin. The data suggest that some of these residues can be assigned to the C-terminal tail suggested to be mobile. At least some of the signals assigned to the C-terminus disappear in the presence of MAP7 indicating a direct interaction with this protein.

With a successful isotope labelling procedure the paper presents an important step towards further, more detailed successful NMR-studies on microtubule-protein interaction. However, the actually biological information obtained from this study is limited, since the assignment of the resonances has not been performed. These data should be published and made available to the scientific community but the reviewer is not really sure if at the present state the biological results are important enough to get published in Nature Communications.

Point by point response:

Reviewer #1 (Remarks to the Author):

[1] The authors successfully generated isotope-labelled tubulin and assessed the binding of microtubules and MAP7 by solid state NMR. They managed to detect differences in the C-terminal portion of tubulin in the absence and presence of MAP7 using ssNMR, indicating an interaction of MAP7 through tubulin C-terminal tail.

It is nice to see that this type of experiment provides insights, and potentially applicable to assess binding of unstructured protein regions to interaction partners.

I am however concerned if there is enough novelty in this study and I am afraid that it would be better suited for a more specialized journal.

Response:

We thank the reviewer for her/his comments. As discussed in our earlier submission, our study for the first time allows to directly examine the structural and dynamical properties of microtubules at atomic level by NMR.

We have expanded the introduction (pages 3 and 4) to stress the preparative and technical difficulties that so far made such NMR experiments impossible. In addition, textual additions on pages 6 and 7-10 and additional references explain in further detail how we can infer dynamic and structural information from our NMR data.

On the other hand, we have stressed throughout our revised manuscript (Abstract, Introduction, pages 3&4 and discussion, pages 10&11) that our work for the first time elucidates the structural and dynamical properties of tubulin tails that are widely recognized as a modulator of microtubular function but which have remained elusive in all available X-ray or EM structures.

For the same reason, we have also provided new data (pages 7-9, new figure 4) that underline the novelty and importance of such studies: We utilized our new approach to study the details of binding to microtubules of another protein domain – the CKK domain of CAMSAP1, a microtubule minus-end binding protein. Our previous work showed that CKK binds in the groove between two protofilaments, and requires the C-terminal tubulin tails for efficient binding. However, the mode of the engagement of tubulin tails with CKK was not deciphered, because the tails were not visible in Cryo-EM studies (ref. 15 and 19 in the main text).

In our NMR studies, we observe that the dynamics of both tubulin C-terminal tails are rapidly modulated (in the nano- to microsecond scale) by the bound CKK domain. In contrast, MAP7 binding to MTs is characterized by tubulin C-terminal tails that exchange between bound (major state) and free (minor state) conformations on a much slower (millisecond or slower) time scale.

Taken together, these findings suggest that MAP-tubulin tail interactions can take place over a remarkably wide range of time scales and, as we now write in

the discussion section (page 10 & 11), may at least in part be determined by characteristics of the MAP binding site and the structural and dynamical properties of the microtubule-associated protein itself. As we further indicated in our revised version, future work using our NMR approach will be ideally suited to obtain additional insight into these dependencies.

Specific concerns are as follows:

[2] *The authors can assign peaks for glutamate, glycine and alanine for the interaction at the C-terminal tail of tubulin. However it is not clear if this is really a generally applicable observation for other MAPs.*

Response:

As discussed under [1], our goal in the current paper was to directly probe the dynamics of tubulin tails before and after addition of MAPs. We find that this interaction mode can take place over a wide range of time scales. As we already wrote in our original submission (now on page 11), it is also possible to obtain NMR assignments for dynamic regions of MAPs provided that they are produced in [¹³C,¹⁵N] labeled form. For example, we are planning to conduct such studies in the future on CKK and MAP7 proteins.

[3] *Can the authors do a control experiment to see the interaction to other proteins that are supposed to bind to the tubulin C-terminus, such as Tau and CAP-Gly domain-containing proteins?*

Response:

We thank the reviewer for suggesting additional control experiments. As a positive control, we have used the human CAMSAP1 CKK domain N1492A mutant in complex with ¹³C, ¹⁵N labeled microtubules, which has been showed to display a more homogeneous and stronger binding compared to the wild-type CKK domain in our previous studies that interact the tubulin C-terminal tails. We performed the same J-based type of experiments and confirmed that the tubulin tails interact with CKK and the dynamical properties were affected by the binding (see also points [1] and [2] above).

[4] *Further it seems necessary to do a negative control experiment with proteins that do not primarily interact with the C-terminal tail. For example, kinesin, as the authors discussed in the manuscript.*

Response:

We thank the reviewer for this suggestion. To our knowledge, previous studies have shown that the tubulin tails affect kinesin binding (Wang and Sheetz, Biophys J 78 (2000) 1955, Lessard et al., JBC 294 (2019) 6353) and many MAPs are known to interact with tubulin tails (ref. 40 in our revised paper). Therefore, we currently cannot think of a suitable negative control experiment. Even if tubulin tails are not strictly necessary for binding of a certain protein to microtubules, they will likely still engage with it, and this engagement will be detectable by NMR. In fact, we think that MAP7 might represent such a case, as our NMR data indicate that the tails can bind and unbind from MAP7.

[5] The model is rather confusing and it does not add any new information learned from this study. The problem is that the model is based on a previously published observation that MAP7 and Tau compete and not by the data from this study.

Response:

We thank for the reviewer for this comment. For the sake of clarity, we have removed the model and concentrate on pages 9 and 10 on our NMR results and their interpretation in our new Figure 5 (revised figure 4 from earlier submission). We kept the secondary structure prediction MAP7 (including the exact amino-acid sequence of the putative termini) in Figure 5a because it reveals interesting differences to the case of the CKK termini (which is now included in Figure 4c). This aspect is considered in our revised discussion section (page 11).

Minor:

[6] In the discussion the authors mention the MAP7 binding competes with tau but not with kinesin-1. This should be more carefully mentioned, as the authors only used the MTBD of MAP7 and the entire MAP7 is much larger. Competitive binding does not only happen through the small binding sites for the tubulin tails but in a larger context such as electrostatic interactions and steric hindrance.

Response:

We agree with the reviewer. A recent study showed that the intrinsically disordered region also contributes to the binding of microtubules (Tymanskyj et al, elife 7 (2018) e36374). Besides tubulin tails and the H11-H12 helices, electrostatic interactions and steric hindrance may indeed contribute to the competitive binding and we plan to further investigate these aspects in future work.

[7] Which R and K residues in MAP7 are in close proximity to H12 helix?

Response:

In line with our comments in [5], we have removed this model to clarify our findings. We agree with the reviewer that further work will be needed to resolve these residues.

[8] It looks like the binding of MAP7 to microtubules causes bundling (S.fig 1A and B). The authors should comment if there is any effect in the ssNMR measurement.

Response:

In principle, the bundling of MTs does not affect the structure and the dynamics of each individual microtubule. Therefore, this aspect should not give influence our ssNMR measurements. We have clarified this in the revised manuscript on page 4.

Reviewer #2 (Remarks to the Author):

[1] The authors report a solid-state NMR study on the interaction of microtubules (MTs) with MT-associated proteins (MAPs), which are regulated by GDP/GTP binding. MAPs are intrinsically disordered and seem to interact with the dynamic MT surface. The molecular details are unknown.

The authors have prepared ^{13}C - ^{15}N -MT from human HELA cells. They are functional and well-folded, which is a remarkable achievement. Their protocol relies on the use of HELA S3 cells and the application of polymerization/depolymerization cycles. MT was stabilized by taxol.

Isotope labelling of MT enabled the recording of first ^{13}C - ^{13}C MAS-NMR spectra. Microscopic controls before and after MAS sample spinning confirmed sample integrity.

A qualitative comparison of the dipolar ^{13}C spectra with empirical chemical predictions revealed a good match for the Ser/Thr Ca/Cb region and lack of residues in the Ala Ca/Cb area. The latter is explained by loop motions.

Furthermore, bound GDP and GTP could be detected by ^{31}P -cross polarization. Here, well-resolved spectra could be obtained indicating good sample homogeneity.

Scalar-based experiments confirmed high mobility and disorder of residues with the alpha- and beta-tubulin tails. Upon binding of MAP7, the intensity of some peaks is reduced, which supports that the corresponding residues are involved in binding.

Combining these observations with the MAP7 secondary structure and known biochemical observations regarding protein tau and kinesin-1 binding, a cartoon as derived in which the potential location of MAP7 is indicated. Overall, this is a solid study, which reports important progress. It is still on a very qualitative / proof-of-concept stage but the possibility to produce such samples enables a range of novel experiments as the authors correctly point out in their outlook.

Response:

We thank the reviewer for these comments.

I only have a couple of questions/suggestions:

[2] - The authors also describe the preparation of MT without taxol and refer to Fig. 1c, but it was not clear to me to which conclusion these data lead in the context of this paper. Additional explanations would be helpful.

Response:

We thank the reviewer for this comment. We have added an additional

statement in the Discussion section (page 10) that clarifies this aspect: Our ability to produce labeled tubulin will allow us to also study, for example, microtubule nucleation and the effects of MAPs on this process, which involves soluble tubulin, as opposed to taxol-stabilized microtubules.

[3] - It appears that the lipid and P_alpha' peaks in Fig. 2b are broader than the other resonances. Did the authors record Hahn-echo experiments or spectra at different CP contact times to further characterize the dynamics of the bound species?

Response:

We thank the reviewer for her/his comments. In the current context we only recorded the experiment using a CP contact time that was optimal to maximize signal intensity. However, the suggested experiments are interesting and can be further studied using unlabeled (or commercially available) tubulin and MTs produced using our protocol.

[4] - The model in Fig. 4 is more a cartoon. Would it be possible, already based on these data, to derive a better model for example based on a docking/MD approach?

Response:

In line with our comments to reviewer 1, point [5], we have decided to remove the model from our current manuscript. These aspects are subject of ongoing studies in our laboratory.

Reviewer #3 (Remarks to the Author):

The manuscript by Luo et al. reports the observation of the interaction of microtubules associated protein MAP7 with human microtubules using solid-state NMR spectroscopy.

There are already a number of studies of isotope labeled MAP proteins with microtubules, but the inverse experiment, taking isotope enriched tubulin could not be performed since the expression yield of labeled tubulin was not sufficient for NMR studies. The authors describe the application of a labeling procedure in HELA cells to tubulin that has been published earlier by them for EGFR. In fact they obtained natively ¹⁵N, ¹³C enriched folded tubulin. They show solid state NMR spectra of tubulin. This is an important step for further solid state studies of microtubules. By scalar-based (J-based) ssNMR experiments they can identify some signals from mobile segments of tubulin. The data suggest that some of these residues can be assigned to the C-terminal tail suggested to be mobile. At least some of the signals assigned to the C-terminus disappear in the presence of MAP7 indicating a direct interaction with this protein.

With a successful isotope labelling procedure the paper presents an important step towards further, more detailed successful NMR-studies on microtubule-protein interaction.

[1] However, the actually biological information obtained from this study is limited, since the assignment of the resonances has not been performed. These data should be published and made available to the scientific community but the reviewer is not really sure if at the present state the biological results are important enough to get published in Nature Communications.

Response:

We thank the reviewer for her/his comments.

In our revised paper, we have extended the discussion on our NMR analysis of the C-terminals of tubulin (pages 6 and 7). We agree with the reviewer that the NMR assignments are not complete. However, the presented 2D and 3D NMR data allowed us to clearly identify amino-acid stretches which are characteristic for α - and β -tubulin tails. Moreover the amino-acid interpretation of our data is in very good agreement with earlier work (new ref. 40), where solution-state NMR experiments were conducted on soluble peptides that contain the tails of α -tubulin. With these results we also could tentatively assign Tyr 451. As we had mentioned in the Discussion section, further assignments of the C-terminal tails may be possible in the future by using scalar-based 3D proton detected experiments or employing tailored isotope labeling.

Already with the current set of tentative NMR assignments, we were able to for the first time study how C-terminal tubulin tails interact with different microtubule-associated proteins (MAPs) across different time scales. As we further expanded in the revised manuscript, these tubulin tail – MAP interactions are critically involved in microtubule organization and function, and recent evidence implicates them also in compartmentalization of MAPs on microtubules (see page 11 in the extended discussion and the cited refs. 46, 48 & 49). Again, our approach could provide a powerful method to dissect the structural and dynamical aspects of these interactions in a functional and disease related framework at atomic level. We also refer the interested reader to our recent study (new ref. 38) which demonstrates the potential of such experiments.

Reviewers' Comments:

Reviewer #1:

Remarks to the Author:

The authors sufficiently addressed the concerns and I am happy to support the publication of this study in Nature Communications.

Minor:

On page8 (in red), results and interpretation are mixed. It would be good if the authors look through the paragraph once again more carefully and try to make it clear.

Reviewer #2:

Remarks to the Author:

The authors have fully addressed all of my comments. I have no further suggestions.

Point by point response:

Reviewer #1 (Remarks to the Author):

[1] The authors sufficiently addressed the concerns and I am happy to support the publication of this study in Nature Communications.

Minor:

On page8 (in red), results and interpretation are mixed. It would be good if the authors look through the paragraph once again more carefully and try to make it clear.

Response:

We thank the reviewer for her/his comment. To further clarify page 8, we extended a sentence on page 7 which now emphasizes that our observed NMR changes relate to the arginine residues of MTs as well as to both the C-terminal residues of α -tubulin and of β -tubulin. Subsequently, we added a sentence on page 7 and slightly regrouped the text on page 8 to further make clear that these three aspects are discussed in different paragraphs on pages 7 and 8.

Reviewer #2 (Remarks to the Author):

The authors have fully addressed all of my comments. I have no further suggestions.

Response:

We thank the reviewer for this statement.